# Sensor-Driven Real-Time Recognition of Basketball Goal States Using IMU and Deep Learning

**DOI:** 10.3390/s25123709

**Published:** 2025-06-13

**Authors:** Jiajin Zhang, Rong Guo, Yan Zhu, Yonglin Che, Yucheng Zeng, Lin Yu, Ziqiong Yang, Jianke Yang

**Affiliations:** 1College of Big Data, Yunnan Agricultural University, Kunming 650201, China; zjjclc@ynau.edu.cn (J.Z.); 2023210516@stu.ynau.edu.cn (R.G.); 2024240737@stu.ynau.edu.cn (Y.C.); 2024240731@stu.ynau.edu.cn (Y.Z.); 2024210530@stu.ynau.edu.cn (L.Y.); 2002033@ynau.edu.cn (Z.Y.); 2Center for Sports Intelligence Innovation and Application, Yunnan Agricultural University, Kunming 650201, China; zhuyan@ynau.edu.cn; 3College of Physical Education, Yunnan Agricultural University, Kunming 650201, China

**Keywords:** real time, IMU sensors, basketball analytics, goal states, deep learning

## Abstract

In recent years, advances in artificial intelligence, machine vision, and the Internet of Things have significantly impacted sports analytics, particularly basketball, where accurate measurement and analysis of player performance have become increasingly important. This study proposes a real-time goal state recognition system based on inertial measurement unit (IMU) sensors, focusing on four shooting scenarios: rebounds, swishes, other shots, and misses. By installing IMU sensors around the basketball net, the system captures real-time data on acceleration, angular velocity, and angular changes to comprehensively analyze the fluency and success rate of shooting execution, utilizing five deep learning models—convolutional neural network (CNN), recurrent neural network (RNN), long short-term memory (LSTM), CNN-LSTM, and CNN-LSTM-Attention—to classify shot types. Experimental results indicate that the CNN-LSTM-Attention model outperformed other models with an accuracy of 87.79% in identifying goal states. This result represents a commanding level of real-time goal state recognition, demonstrating the robustness and efficiency of the model in complex sports environments. This high accuracy not only supports the application of the system in skill analysis and sports performance evaluation but also lays a solid foundation for the development of intelligent basketball training equipment, providing an efficient and practical solution for athletes and coaches.

## 1. Introduction

In recent years, the widespread application of artificial intelligence, machine vision, and the Internet of Things has significantly affected the development of sports [1]. Sports data have become increasingly important. In sports such as basketball, football, tennis, and baseball, data are used for game preparation, improving athlete and team performance, analyzing opponents, and monitoring training progress [2,3,4,5,6]. Action recognition is particularly important in sports analysis. A motion recognition system can provide objective measurements and analysis for sports science, improve the accuracy of sports performance, and evaluate the effectiveness of training plans designed by coaches [7]. Using computer vision and inertial sensor technology, data can be measured and collected, and common motion actions can be automatically recognized through machine learning and deep learning [8].

Shooting plays a crucial role in basketball, and real-time feedback on successful shooting is crucial for both players and coaches [9]. Shooting is the only way to score in basketball games [10]. Basketball shooting refers to all movements that push the basketball towards the basket and is one of the most complex technical movements. The purpose of all tactical applications is to create more and better shooting opportunities and to strive to score through shooting. Previous research has mainly focused on the implementation of wearable devices and the recording of video inputs [11]. However, these methods often overlook information collected from baskets, backboards, and nets. When a basketball is thrown, the ball interacts with the basket, which is manifested as the shaking of the frame and net, caused by rebounds and different shot types. The vibrations of the goal components caused by different types of goals differ [12,13,14,15]. Through extensive literature review and research, each stroke of the ball was found to produce a brief shake of the net until it stabilized in a stationary position [16,17]. The time required to reach a stable position depends on the intensity and smoothness of the vibration. In addition, the success of shooting is highly correlated with jitter data [18]. To demonstrate the connection between goals and goal components, this study proposes a new method that includes an analysis of the basketball, backboards, and nets. By collecting and analyzing data on these factors, a more comprehensive understanding of players’ shooting skills and their impact on shooting success can be acquired. This method considers not only the interaction between the ball and the basket but also the role of the backboards and net in determining the shooting result. The relationship between the backboard, basket, and basketball in the experiment is shown in Figure 1.

By analyzing vibration data, a deeper understanding of the consistency and fluency of player shooting executions can be acquired. Comparing the vibration patterns across players or shot types allows trends to be identified, and using the insights acquired, shooting skills can be improved by developing more informed and effective training programs. Beyond vibration data, additional metrics such as basket position, rebounding height, and net tension provide valuable insights into players’ shooting performance. Combining these metrics with vibration data offers a comprehensive perspective on shooting execution and its impact on success rates. This approach has the potential to significantly enhance basketball players’ analysis of their own skills to improve their on-court performance.

This paper introduces a novel method that utilizes inertial measurement unit (IMU) sensors mounted on a basketball net to detect successful shots and provide players with real-time performance feedback, including smoothness and execution type. This research makes the following contributions:(1)Quantification of goal states: Using IMU sensors, the system captures real-time acceleration, angular velocity, and angular changes during the shooting process, whereby motion features are analyzed across four goal states: rebound, swish, other goal types, and missed shots.(2)Goal state recognition on edge devices: Motion feature data are processed using five deep learning models—convolutional neural network (CNN), recurrent neural network (RNN), long short-term memory (LSTM), CNN-LSTM, and CNN-LSTM-Attention—enabling direct real-time shooting state recognition on edge devices.(3)Visual feedback: Real-time recognition results are integrated into a smartphone application, providing players with an intuitive visualization of the shooting outcomes.

The rest of this paper is organized as follows. In Section 2, related work and current research status are reviewed. Section 3 provides an overview of the system design and hardware platform. In Section 4, deep learning models, including data collection, training, evaluation, and performance analysis, are introduced. In Section 5, the smartphone application is described along with comparisons, limitations, and future directions in relation to existing work. Section 6 provides a summary of key findings and contributions.

## 2. Related Work

### Advances in Basketball Shot Selection and Performance Analysis

Sports analytics continues to present new challenges and opportunities, particularly in predicting outcomes in games like football and basketball. Advances in player tracking technology have significantly improved data collection, expanding the scope and depth of sports data analysis [19]. While numerous sensor-based feedback applications have been developed across various sports, such as diving [20], volleyball [21], golf [22], and skiing [23], many of these systems rely on centralized cloud computing frameworks. This reliance introduces latency, which limits their effectiveness in fast-paced activities like basketball that demand immediate feedback. Moreover, existing real-time feedback systems are often designed primarily for coaches’ external analysis rather than providing instantaneous guidance to players. In contrast, edge computing enables localized data processing near the source, reducing latency and enhancing responsiveness and privacy, making it a promising solution for delivering real-time feedback in dynamic sports environments.

Building on this, recent advances in basketball research have significantly enhanced the understanding of shot selection and performance. Research combining statistical methods, wearable technology, and machine learning has provided new insights into the optimization of offensive strategies and the improvement of individual shooting skills. Ji [24] used image feature extraction and machine learning to analyze basketball shooting motions, thereby providing a framework for studying techniques and performance in professional settings. Similarly, wearable IMUs are increasingly being used to analyze basketball. Acikmese et al. [25] developed an artificial intelligence training expert system that combines sensors and machine learning algorithms to help players improve their techniques. Ma et al. [26] placed an IMU on players’ wrists to identify shooting motions, achieving high accuracy in classifying different types of motions. Cervone et al. [27] extended this work by developing a multi-resolution stochastic process model to predict basketball possession outcomes, providing valuable insights into understanding and optimizing game strategies.

To understand the psychology and physiology of shooting, Gonçalves et al. [28] demonstrated that positive feedback can enhance children’s learning of basketball free throws. Sandholtz et al. [29] and Wu and Bornn [30] studied spatial efficiency and offensive player movements using advanced statistical models to optimize shot selection strategies in professional leagues. Viswanathan et al. conducted anthropometric assessments of young national tournament basketball players and analyzed the physical characteristics associated with playing positions to identify performance-related patterns [31].

The synergy between wearable sensors, machine learning, and image analysis continues to push the boundaries of basketball performance research, with applications ranging from training optimization to in-game decision support. These studies highlight the growing importance of interdisciplinary approaches in advancing basketball analytics and suggest that data-driven sports performance improvement has a bright future.

Liu and Liu proposed a shot recognition system based on real-time basketball images captured using IoT technology, advancing the application of intelligent analysis in basketball performance evaluation [32]. Their method uses IoT-enabled sensors and machine learning algorithms to classify different types of shots, thereby evaluating shot accuracy with enhanced real-time accuracy. However, few studies have focused on using edge computing to enable real-time analysis of basketball shot status data. Edge computing can not only reduce reliance on cloud services but also provide faster feedback by processing data at the source. This is particularly important in fast-paced sports, such as basketball, where split-second decisions can impact performance.

This study builds on these advances by using edge computing to analyze the dynamics of net jitter in real time. The edge-based system minimizes latency and improves real-time performance. By addressing the limitations of existing centralized approaches, this study provides coaches and players with a valuable tool for improving shooting accuracy, paving the way for future intelligent sports training systems. Overall, these studies illustrate the potential of integrating technology and data analytics to improve sports performance. Although advances in basketball shot selection and performance analysis are promising, further research is required to effectively leverage the data to improve shooting performance and overall team dynamics. This study aims to enhance basketball analytics by exploring the dynamics of net jitter during a goal, providing coaches and players with valuable insights to improve shooting accuracy.

## 3. System Design

### 3.1. System Overview

Our system was designed to achieve efficient real-time basketball goal-state detection. The system primarily consists of a low-power Arm Cortex^®^-M4F STM32L476 microcontroller unit (MCU) (STMicroelectronics, Geneva, Switzerland) and a STEVAL-STLCS01V1 (STMicroelectronics, Geneva, Switzerland) inertial measurement unit within the Sensor Tile module. The LSM6DSM (STMicroelectronics, Geneva, Switzerland) sensor integrated into the STEVAL-STLCS01V1 captures the 3D acceleration and angular velocity changes, providing essential feature data for identifying different goal states.

As shown in Figure 2, the system is divided into four key functional modules: data acquisition, signal filtering, signal preprocessing, and deep learning model recognition. When the basketball enters the hoop and interacts with the net, the IMU sensor collects real-time vibration data from the net. The data are transmitted to the Arm Cortex^®^-M4F STM32L476 microcontroller for real-time edge processing.

Upon detecting net vibrations, the system applies a moving average filter to remove noise from the IMU signals, thereby improving signal stability. Next, a sliding window technique is used to segment the signal into multiple intervals, each corresponding to a different goal state. Key features are then extracted from these segmented signals and used as inputs for deep learning models to classify and recognize the goal states.

Deep learning models enable the system to accurately classify data into four distinct goal states. The recognition results are immediately transmitted to external devices such as computers or smartphones via Bluetooth wireless transmission. This ensures seamless data integration and supports remote analysis.

### 3.2. IMU Sensors

The experiment uses the Sensor Tile development kit STEVAL-STLKT01V1, which integrates Sensor Tile modules, expansion boards, and portable extension boards to streamline the development process and provide comprehensive hardware support. STEVAL-STLKT01V1 includes several components: a microcontroller, a microphone, a Bluetooth module, and an IMU that integrates accelerometers, gyroscopes, and magnetometers. In this experiment, we primarily utilized the LSM6DSM inertial sensor embedded in the STEVAL-STLCS01V1 board to monitor and analyze the motion states, which helps identify the basketball goal status.

To achieve the wireless transmission of sensor data, the STLCS01V1 module was soldered onto the STLCR01V1 (STMicroelectronics, Geneva, Switzerland) Bluetooth module. Soldering ensures stable signal transmission, minimizes interference and data loss, and improves the reliability and real-time performance of the system. The soldering connection method not only strengthens the physical and electrical connections between modules but also maintains the integrity of the collected data.

Figure 3 illustrates the soldering method used for the STLCS01V1 and STLCR01V1. The key parameter information of the STEVAL-STLCS01V1 core system and STLCR01V1 Bluetooth module is summarized in Table 1 and Table 2, respectively. Additionally, Table 3 presents the key specifications of the LSM6DSM inertial sensor used in the experiment. These components formed the backbone of the experiment, providing robust hardware capabilities for motion analysis and real-time data transmission.

### 3.3. Edge Computing Unit

The present study uses the STM32L476 MCU as the edge device responsible for the acquisition, preprocessing, reasoning, and recognition of IMU sensor data with the support of edge deep learning algorithms.

The STM32L476 MCU is a mid-range model of the STM32L4 series, characterized by its ultralow-power single-chip architecture, which is suitable for edge computing applications. It runs on an ARM Cortex M4F core with a frequency of 80 MHz, integrating a hardware floating-point unit (FPU) and a digital signal processing (DSP) instruction set, and has 1024 kB program memory (flash) and 128 kB RAM. To develop the STM32L476 microcontroller, we used the expansion board in the Sensor Tile development kit STEVAL-STLKT01V1, which supports a USB connection and power supply. ST-Link V2 (STMicroelectronics, Geneva, Switzerland) on the NucleoSTM32L476 (STMicroelectronics, Geneva, Switzerland) board was used for device burning and debugging operations to ensure the efficiency and stability of system development and debugging. The entire setup is shown in Figure 4.

The STM32L476 microcontroller runs the API of the IMU sensors, which can control the operation of the sensor, configure the parameters, and read measurement data. STMicroelectronics X-Cube-AI (STMicroelectronics, Geneva, Switzerland) artificial intelligence kit was used to load the deep learning model trained on the PC server into STM32L476 and perform benchmark tests to achieve efficient real-time reasoning.

## 4. Experimental Setup

During the writing of this manuscript, GPT-4o (OpenAI, San Francisco, CA, USA, 2024) was used to enhance the language quality, including sentence structure optimization, vocabulary refinement, and logical organization. In particular, it was employed to polish the English description of the model architecture based on a Chinese draft and associated figures. All AI-assisted content was carefully reviewed and edited by the authors to ensure accuracy and consistency with the research. All experimental data were obtained from real experiments conducted in compliance with strict scientific procedures.

This flowchart shows the processing flow of goal state recognition, including four steps: data collection, preprocessing, deep learning model processing, and recognition output. After data collection, preprocessing, such as noise reduction, normalization, and segmentation, was performed, analyzing the results using a deep learning model to identify the goal types (see Figure 5).

### 4.1. Data Collection

To fully analyze the basketball shooting performance, sensor and video data were collected during the experiment.

Sensor data collection: The data were collected at the Sports Center of the Yunnan Agricultural University in Kunming, Yunnan Province, China. The participants included 30 individuals: 10 male basketball players, 10 female basketball players, and 10 amateur basketball enthusiasts, all performing various types of shots within the three-point line. The participants performed 30 rebound shots in the first round and 30 non-rebound shots (20 empty shots and 10 non-empty) in the second round. Each participant performed 30 rebound shots in the first round and 30 non-rebound shots in the second round, which included 20 empty shots and 10 non-empty shots. In total, 900 shooting samples were collected (30 participants × 60 shots). The basketball backboard and basket used in the experiments were of standard size.

The core system of the Sensor Tile development board (including the support board and battery) was welded and encapsulated in a plastic housing. The assembly was firmly placed outside the lower end of the basketball net. The equipment was firmly fixed in the same position throughout the data collection phase to ensure consistency and accuracy. The experimental setup is shown in Figure 6A. The sensors continuously recorded acceleration and angular velocity values along the x-, y-, and z-axes, capturing the key moments of the basketball’s interaction with the goal.

Video capture and annotation: To complement the sensor data, video recordings of the shooting process were captured for data labeling and verification. These videos were used to define and manually annotate the exact moments of the basketball interaction with the goal, ensuring alignment with the timestamps of the sensor data. Figure 6B shows a specific moment captured during the shooting process. Through video analysis, the researchers observed the trajectory of the ball interacting with the goal components. This visual information provided key insights for labeling and interpreting the sensor data, enabling the accurate identification of key events, such as the ball hitting the net or the basket.

Combining sensor data with video analysis provides a more complete understanding of basketball shooting performance. The plotted sensor data show clear fluctuations corresponding to different goal states, such as empty shots, rebounds, and missed shots. The collected data, especially the moment when the ball hits the goal, were preprocessed and stored for further analysis.

### 4.2. Data Preprocessing

In this experiment, data cleaning was a key step in ensuring data quality and reliability, encompassing noise removal, sliding window segmentation, time synchronization, and goal status confirmation.

First, high-frequency noise in the raw sensor signals was smoothed using a sliding average (moving average) filter, which preserves overall trends while eliminating random spikes. For each sensor axis x, the denoised value at time i is computed as follows:(1)$$\tilde{x_{i}}=\frac{1}{k}\sum_{j=i-\lfloor k/2\rfloor }^{i+\lfloor k/2\rfloor }x_{j}$$
where k is the window size, empirically chosen between 5 and 11 based on noise levels observed in pretests [33].

Next, the time-series data were segmented using a fixed-length sliding window of 150 samples (equivalent to 3 s at a 50 Hz sampling rate), with a 50% overlap (75 samples). This configuration ensured that each shooting motion—typically lasting 2 to 3 s—would be fully contained within at least one segment. Within each segment, the average vector magnitude of acceleration was calculated as follows [34]:(2)$$a_{avg}=\frac{1}{N}\sum_{i=1}^{N}\sqrt{{a^{2}}_{x,i}+{a^{2}}_{y,i}+{a^{2}}_{z,i}}$$

If $a_{avg}>70$, the segment was flagged as potentially containing a shooting action. This threshold was determined through empirical analysis of labeled data. The maximum acceleration within such a segment was designated as the “impact point,” typically corresponding to the moment of ball release or wrist snap [35].

To synchronize sensor data with actual movement events, video recordings were used as references. Peaks and valleys in accelerometer (ax,ay,az) and gyroscope (gx,gy,gz) data along all three axes were compared with video frames to identify three key time points:

Begin point: onset of movement initiation,

Impact point: peak dynamic activity,

End point: completion of shooting or follow-through.

These points were aligned with corresponding video frames (see Figure 7), ensuring accurate temporal mapping between the sensor signals and physical movements [36].

Finally, the goal status—hollow ball, rebound ball, or no goal—was determined by integrating time-aligned video annotations with post-impact sensor characteristics, such as sustained vibration in az or notable angular velocity in gy. Sliding window features also contributed to resolving ambiguous cases and refining the classification.

This integrated approach—leveraging both sensor data and video analysis—offers a comprehensive understanding of basketball shooting actions. As illustrated in Figure 7A–D, the extracted time-series trends with marked key events validate the robustness of our preprocessing pipeline and provide high-quality inputs for subsequent model training and action recognition tasks.

#### Labeling of Collected Data

During data collection, with the sensor fixed on the basketball net, we annotated the shooting results by analyzing the vibration pattern of the basketball net and video recordings. Specific categories include shooting miss (SM), that is, the shot did not enter the basket, with the sensor only recording a slight fluctuation and rebound (RB), that is, the basketball hits the board and then enters the basket, with the sensor capturing a significantly different vibration feature compared with the hollow ball (HB). For basketballs going directly into the basket, the sensor captured a fast and smooth vibration pattern, and others (OT) included multiple touches or complex scoring situations. By combining sensor and video data, we established a target classification system based on the motion characteristics of the basketball net. Details of the classification definitions are listed in the following Table 4:

### 4.3. Model Selection

Unlike traditional machine learning techniques, deep learning methods can autonomously identify and extract relevant features directly from raw data, thereby eliminating the need for manually designed features. This approach has attracted considerable attention in the field of motion recognition. In this study, we investigate five deep learning models—CNN, RNN, LSTM, CNN-LSTM, and CNN-LSTM-Attention—to classify four types of basketball goal states: Rebound, Hollow Ball, Other, and Shooting Miss. All models are trained on the same dataset, where each sample is represented as a multivariate time series with a shape of (6, 6), corresponding to six time steps and six features per step. These features include the three-axis accelerometer data (X, Y, Z) and three-axis gyroscope data (X, Y, Z) captured during each basketball shot. A total of 150 samples were collected and manually labeled according to the goal type. This consistent input format ensures a fair comparison across different model architectures.

#### 4.3.1. CNN Model

In image recognition tasks, CNN is widely used for its ability to automatically learn and identify significant features from raw image data. Alternating the convolutional and pooling layers allows capture of local features while reducing computation, effectively mitigating overfitting risks, as demonstrated in previous studies [37]. In this experiment, a CNN was used to analyze basketball net vibration data, capturing both the spatial and temporal features critical for accurate shot classification. The architecture followed that of a conventional CNN approach, as described in similar studies on sports data analysis [38], illustrating the effectiveness of CNN in tasks that require high-dimensional spatial feature extraction [39].

The CNN model used in this experiment was specifically designed to recognize shooting actions. Deep learning techniques were leveraged to automatically extract and learn critical features from input data. The model architecture is shown in Figure 8. The process begins with the input layer, which accepts grayscale image data with a shape of (None, 6, 6, 1), where “None” represents the batch size. The input is passed to the first convolutional layer (Conv2D), which applies 32 filters of size 3 × 3, generating a feature map of shape (None, 4, 4, 32). This feature map is then passed through a MaxPooling2D layer, which performs downsampling and reduces the spatial dimensions to (None, 2, 2, 32), lowering computational cost and helping to prevent overfitting. A dropout layer follows, with the same output shape.

Next, the data are processed by a second convolutional layer with 64 filters of size 3 × 3, producing an output of shape (None, 2, 2, 64). Another MaxPooling2D layer is applied, further reducing the spatial size to (None, 1, 1, 64). A subsequent dropout layer continues to mitigate overfitting. The resulting feature maps are then flattened into a one-dimensional vector of shape (None, 64) using a flatten layer. This vector is passed to a fully connected (Dense) layer with 128 units, followed by an activation function to introduce non-linearity. A second dropout layer is applied before the final dense layer, which has 4 output units corresponding to the four target classes. This final layer produces output of shape (None, 4), representing the predicted probabilities for each class.

By integrating convolutional, pooling, dropout, and fully connected layers, this CNN model effectively maps raw image data to classification outputs, demonstrating strong capability in image-based action recognition tasks.

#### 4.3.2. RNN Model

The RNN is particularly well-suited for time-series analysis because of its capability to capture long-term dependencies, which is essential for predicting shot outcomes based on net vibration data. Peterson et al. and Wang et al. demonstrated the effectiveness of RNN in motion analysis, showcasing their applicability in sports data tasks involving temporal dynamics [40,41]. In this experiment, we used a simple RNN layer to capture these dependencies, enabling the model to accurately classify shooting outcomes based on real-time basket movement data.

The structure of the RNN model is illustrated in Figure 9. The model begins with an input layer that accepts time-series data with a shape of (None, 6, 6), where each sample contains a sequence of six time steps with six features per step. The input is first processed by a SimpleRNN layer with 64 units. This layer uses a kernel of shape (6 × 64) and a recurrent kernel of shape (64 × 64), outputting a sequence with shape (None, 6, 64). A dropout layer is then applied to the output to reduce overfitting and improve generalization. The next SimpleRNN layer increases the feature dimensionality to 128, using a kernel of shape (64 × 128) and a recurrent kernel of shape (128 × 128). This is followed by another dropout layer. The sequence then passes through a third SimpleRNN layer with 256 units, which further captures temporal dependencies. This layer is followed by a third dropout layer for regularization. The RNN output is then passed to a fully connected (Dense) layer with 128 units and an activation function, followed by a dropout layer. Finally, the model outputs predictions through another dense layer with 4 units, corresponding to the four classification categories. The output layer uses a softmax activation function to generate class probabilities, while all intermediate layers apply the ReLU activation function.

This multi-layer RNN architecture, enhanced with dropout and dense layers, effectively captures temporal dynamics in the input data and maps them to classification outputs, making it well-suited for tasks involving sequential patterns in sports action recognition.

#### 4.3.3. LSTM Model

LSTM networks are variants of RNN, optimized to retain or forget specific information over time, thereby addressing the limitations of traditional RNN in handling long sequences [42]. In sports analysis, LSTM has shown significant potential for capturing motion sequences and predicting actions based on time series data [43]. In this study, LSTM is applied to basketball net movement data, allowing the model to process shooting dynamics and classify shooting results with increased accuracy. Studies by Fok et al. confirmed the effectiveness of LSTM in handling complex sequential datasets typical of sports analysis tasks [44].

In this experiment, the LSTM model was applied to basketball goal state recognition, enabling the system to accurately process and classify goal states. The model structure is shown in Figure 10. The model begins with an input layer, “lstm_input”, which receives sequential data of shape (None, 6, 6), where each sample contains six time steps with six features per step. The first LSTM layer processes this input and maps it to a higher-dimensional space using a kernel of shape (6 × 256) and a recurrent kernel of shape (64 × 256), producing an output of shape (None, 6, 256). To prevent overfitting and improve generalization, a dropout layer is applied to the output of this LSTM layer. The subsequent LSTM layer increases the output dimensionality to 1024 units, with a kernel of shape (128 × 1024) and a recurrent kernel of shape (256 × 1024). A second dropout layer follows this to enhance regularization. The third LSTM layer further processes the sequence using 512 units and receives input from the previous layer, using a recurrent kernel of shape (128 × 512). This layer captures deeper temporal dependencies within the data and is followed by another dropout layer. The output is then passed to a fully connected (Dense) layer with 128 units, followed by a dropout layer for additional regularization. Finally, the processed feature representation is fed into another dense layer with 4 output units. This final layer uses a softmax activation function to produce a four-dimensional output vector, representing the predicted class probabilities for each goal state category.

ReLU activation functions are applied throughout the hidden layers, except for the output layer. This deep LSTM architecture effectively captures both short- and long-term temporal features and provides robust classification performance for sequential sports data.

#### 4.3.4. CNN-LSTM Model

Combining CNN with LSTM can exploit both spatial and temporal features, allowing a more comprehensive analysis of sports actions involving complex movements and sequence dependencies. This CNN-LSTM model architecture was drawn from prior studies, where the fusion of CNN and LSTM significantly improved classification accuracy in time-dependent recognition tasks [45,46]. By integrating a CNN for spatial feature extraction and an LSTM for temporal feature processing, this hybrid approach provides an improved understanding of complex goal states. Such architectures have been noted for their robust performance in motion sequence recognition [47]. Based on the architecture illustrated in Figure 11, the model begins with the input layer named “conv2d_6_input”, which accepts inputs of shape (None, 6, 6, 1), representing a batch of grayscale images with spatial dimensions of 6 × 6 pixels. The input is first processed by a Conv2D layer that uses 3 × 3 kernels, a stride of 1, and 16 filters. This layer outputs a tensor of shape (None, 6, 6, 16), maintaining the spatial dimensions through appropriate padding while increasing the depth to extract 16 feature maps.

Subsequently, a MaxPooling2D layer is applied to downsample the feature maps, typically reducing the spatial resolution to lower computational complexity and highlight dominant features. The resulting tensor is then passed through a reshape layer, which transforms the output into a shape of (None, 6, 64). This reshaping step prepares the data for sequential modeling by organizing it into a format where each of the 6 time steps contains a 64-dimensional feature vector.

The reshaped data are then fed into an LSTM layer with 64 units, which is capable of learning temporal dependencies across the 6-step sequence. The LSTM layer processes the input sequence and outputs a fixed-size feature vector of shape (None, 50). To mitigate overfitting, a dropout layer is introduced next, which randomly deactivates a portion of the neurons during training.

Finally, the output is passed through a dense (fully connected) layer with an output shape of (None, 4), which serves as the classification layer, assigning each input sample to one of four predefined categories. This model architecture effectively combines convolutional feature extraction, temporal sequence modeling using LSTM, and final decision-making through a dense classification layer, providing a comprehensive pipeline for tasks involving both spatial and temporal data characteristics.

#### 4.3.5. CNN-LSTM-Attention Model

The CNN-LSTM-Attention model enhances recognition accuracy by introducing an attention mechanism that prioritizes critical information, making it ideal for recognizing complex patterns in sports data. The use of attention allows the model to focus on pivotal shot features during the scoring process, aligning with recent findings on the effectiveness of attention in refining sequence-based classifications [48]. By combining the CNN, LSTM, and Attention layers, this model capitalizes on spatial, temporal, and feature-specific importance, ultimately achieving improved prediction accuracy for basketball goal states. Recent studies confirm the utility of attention layers in sports applications that require nuanced feature emphasis and time-dependent pattern recognition [49].

The architecture of the CNN-LSTM-Attention model is illustrated in Figure 12; the CNN-LSTM-Attention model begins with the input layer “input_14”, which accepts input data of shape (None, 6, 6, 1), indicating batches of grayscale images with dimensions 6 × 6 pixels. The input is processed by a Conv2D layer with a kernel size of 1 × 3, a stride of 1, and 64 filters, producing an output of shape (None, 6, 6, 64). This layer extracts local spatial features while preserving the input’s spatial dimensions through padding. The output then passes through a MaxPooling2D layer, which performs downsampling to reduce spatial dimensions, enhancing feature robustness and reducing computational load. A subsequent reshape layer transforms the tensor into shape (None, 6, 384), preparing it for sequence modeling by treating the data as a six-step sequence with 384 features per step. A dropout layer is then applied to reduce the risk of overfitting by randomly setting a fraction of input units to zero during training. This is followed by a bidirectional LSTM layer, which processes the sequence both forward and backward to capture past and future context. This results in an output of shape (None, 6, 100), encapsulating temporal dependencies in both directions. Parallel to the LSTM pathway, the output is also processed through a MultiHeadAttention mechanism, which allows the model to focus on different positions of the input simultaneously and extract complex interdependencies. Prior to this, layer normalization is applied to stabilize and accelerate training by normalizing activations. Another dropout layer follows to further regularize the network. The attention output is then added to the original input via an add layer, forming a residual connection that improves gradient flow during backpropagation. Subsequently, the data are passed through a dense layer with 100 units, followed by another dropout, layer normalization, and add layers. These components form a feed-forward block with residual connections, which further refines the extracted features. Afterward, the outputs from the bidirectional LSTM and the attention path are combined using another add operation. The combined tensor is then flattened into a one-dimensional vector through a flatten layer, resulting in shape (None, 600). Finally, this vector is passed into a dense layer with 4 output units, producing the final classification output of shape (None, 4).

In summary, this architecture combines convolutional layers for local feature extraction, bidirectional LSTM for sequence modeling, multi-head attention for capturing global dependencies, and residual connections for stable deep learning. The integration of these components forms a robust pipeline for precise and generalized classification tasks.

#### 4.3.6. Training and Validation

In this section, we provide a deeper examination of the training results for the five models. All four models were compiled with the categorical cross-entropy loss function. The learning rate was set to 0.001, and an efficient adaptive moment estimation (Adam) was used as the optimizer for training, with the number of iterations reaching approximately 100. This carefully designed training scheme ensured that the performance of the model was maximized, thereby laying a solid foundation for subsequent analyses. Training was performed using a computer equipped with an NVIDIA RTX 3080, 10 GB GDDR6X memory, 8704 CUDA cores, 2.80 GHz Intel Core i7-7700 HQ, and 32 GB RAM. The Python (version 3.10; https://www.python.org, accessed 15 December 2024) programming language was used for training and classification. All five deep learning models were trained using the same training, validation, and test datasets to obtain a fair comparison of training performance.

Subsequently, as shown in Figure 13, we performed 8-bit post-training quantization (PTQ) and ported the post-training deep learning model based on STM32 X_CUBE. AI. At the expense of a small accuracy loss, model parameters were compressed, reducing model memory usage and complexity, thereby increasing model execution time. Finally, the quantitative model was converted into a C file, containing the model structure, weight parameters, and calling interface, which was deployed to the STM32L476 microcontroller to recognize shooting and scoring events, as well as facial expressions.

Regarding the division of the training and test sets, we randomly selected 70% of the dataset for training and the remaining 30% for testing. In addition, to account for the potential impact of random factors on the network training process, the network training steps were repeated seven times, using the average results to verify performance. Figure 14A,B show validation loss and accuracy for the five models, respectively. The validation dataset is an unknown dataset. Therefore, the validation loss graph can also be used to observe training progress. As shown in Figure 14A,B, the five models performed differently in terms of loss decline and prediction performance. Although the most basic CNN model effectively extracted spatial features, the decline in loss was gradual, reaching a high final value because of its limited ability to process time-series information and poor performance. The RNN model provided an improvement in processing time dependencies; however, its accuracy was low and fluctuated significantly, and its final loss was also high, making it difficult to cope with complex long time-series problems. In contrast, the LSTM model performed well in capturing long- and short-term dependencies, with a fast and stable decline in loss and significantly higher accuracy than RNN. However, the LSTM is insufficient for processing spatial features. The CNN-LSTM model significantly accelerates the decline in loss by combining spatial feature extraction and temporal dependency modeling, with a lower final loss and better performance than a single model. The best-performing model was the CNN-LSTM-Attention model, which not only combines the advantages of CNN and LSTM but also highlights key features through the attention mechanism, significantly improving prediction accuracy, exhibiting the fastest decline in loss and the lowest final loss value, with fewer fluctuations during the training process. Therefore, the CNN-LSTM-Attention model performed the best when processing complex spatiotemporal data and was the best choice for this prediction task.

To better contextualize the performance of our proposed models, a comparative analysis with existing IMU-based activity recognition studies is presented in Table 5. This table summarizes relevant works that have employed various machine learning and deep learning models for activity recognition tasks using IMU sensors. As shown in Table 5, traditional machine learning methods such as K-NN and Random Forest have achieved moderate accuracy (e.g., 77.56% and 80%, respectively), while more recent studies applying CNN or CNN-LSTM architectures to basketball activity recognition have reported higher performance (up to 85%). Our study achieves the highest accuracy of 87.79%, benefiting from the application of deep learning models (CNN, RNN, and LSTM) and diverse participant data, demonstrating improved capability in handling complex basketball state recognition tasks.

Table 6 shows the final performance of the five models, including key parameters, such as the number of parameters, accuracy, precision, recall, and F1 score. These metrics were used to evaluate the suitability of the models for classification tasks, particularly highlighting their complexity and scale, which are critical when deployed on edge devices. As shown in Table 6, the CNN model had an average accuracy of 64.96%, 93,184 parameters, a recall of 61.03%, an accuracy of 55.33%, an F1 score of 54.63%, and a maximum latency of 3 ms. The RNN model had an average accuracy of 56.83%, parameters, recall of 52.58%, accuracy of 51.99%, F1 score of 51.25%, and maximum latency of 2 ms. The LSTM model performed relatively well, with an average accuracy of 68.85%, 69,636 parameters, a recall rate of 67.02%, an accuracy rate of 66.72%, an F1 score of 65.90%, and a maximum delay of 3 ms. The CNN-LSTM model had an average accuracy of 81.14%, 232,680 parameters, a recall rate of 80.47%, an accuracy rate of 82.22%, an F1 score of 80.49%, and a maximum delay of 3 ms. The CNN-LSTM-Attention model performed the best in terms of accuracy, recall rate, accuracy, and F1 score. Although this model has many parameters and the highest delay, all performance indicators far exceed those of the other models; therefore, the CNN-LSTM-Attention model is the optimal algorithm.

Equations (3)–(8) show how to derive accuracy, precision, recall, and F1 score, true positive rate (*TPR*), and false positive rate (*FPR*), where true positive (*TP*), false positive (*FP*), true negative (*TN*), and false negative (*FN*) are the four key metrics for evaluating the performance of the classification models. *TP* indicates the number of positive examples correctly predicted by the model; *FP* indicates the number of negative examples incorrectly predicted as positive by the model; *TN* indicates the number of negative examples correctly predicted by the model; and *FN* indicates the number of positive examples incorrectly predicted as negative by the model. The high and low values of these four parameters reflect the model’s predictive ability and misclassification rate for positive and negative examples, respectively.

*Accuracy* indicates the proportion of samples correctly predicted by the model compared to the total number of samples. The calculation formula is as follows:(3)$$Accuracy=\frac{TP+TN}{TP+TN+FP+FN}$$

*Precision* indicates the proportion of actual positive samples predicted by the model as positive samples. The calculation formula is as follows:(4)$$Precision=\frac{TP}{TP+FP}$$

*Recall* indicates the proportion of all true-positive samples that are correctly predicted as positive by the model. The calculation formula is as follows:(5)$$Recall=\frac{TP}{TP+FN}$$

*F*1 *score* is the harmonic mean of the precision, and recall is used to comprehensively evaluate the performance of the model. The calculation formula is as follows:(6)$$F1\;score=2\times \frac{Precision\times Recall}{Precision+Recall}$$

The receiver operating characteristic (ROC) curve is a graphical tool used to evaluate the performance of classification models in binary classification problems. This reflects the overall performance of the classifier by plotting the changes in *TPR* and *FPR* at different thresholds. Specifically, *TPR* is also known as sensitivity or recall and is calculated as follows:(7)$$TPR=\frac{TP}{TP+FN}$$
with *FPR* calculated as follows:(8)$$FPR=\frac{FP}{FP+TN}$$

The area under the ROC curve (AUC) is an important indicator for measuring the performance of a classifier. The closer the AUC value is to 1, the better the performance of the classifier. In identifying goal states, a good classifier will significantly increase the true positive rate while maintaining a low false positive rate as the threshold changes, causing the ROC curve to be closer to the upper left corner. An AUC close to 1 indicates that the classifier has excellent distinguishing performance for different types of shots and goals. In this experiment, to identify four types of shots and goals, including RB (Rebound), HB (Hollow Ball), OT (Other Types), and SM (Missed Shot), five deep learning algorithms—CNN, RNN, LSTM, CNN-LSTM, and CNN-LSTM-Attention—were applied and compared.

As shown in Figure 15A–E, the CNN-LSTM-Attention model demonstrated the strongest recognition ability in this task. It not only performed extremely well on relatively easy-to-recognize categories such as HB and OT but also achieved a relatively high accuracy on more challenging categories like RB and SM. Therefore, the CNN-LSTM-Attention algorithm was identified as the best choice for the goal states recognition task. This also highlights the practical advantage of integrating attention mechanisms, which can significantly improve model performance in complex time-series and multi-class classification tasks by allowing the model to focus on the most relevant temporal features.

In addition, Figure 16A–E shows the confusion matrices of the four goal states for the five models tested. The most notable misclassification occurred between RB and SM. Specifically, as shown in Figure 15A–C, although the highest recognition rates for HB and OT reached up to 87%, the confusion rate between RB and SM was comparatively higher. This phenomenon can be explained by the nature of the sensor data and the similarity of net vibration signals during these two types of shots. When a rebound occurs, the net vibrates with a certain frequency due to the ball hitting it after bouncing off the rim or backboard. However, when a shot misses and the ball directly strikes the backboard, the net may also exhibit a slight vibration. Because these vibrations can have overlapping frequency characteristics, the sensor sometimes misclassifies the rebound as a missed shot or vice versa.

Furthermore, environmental noise, slight variations in shooter technique, and subtle differences in shot dynamics can add variability to the sensor signals, further complicating classification. These factors make it challenging for the model to perfectly distinguish between RB and SM, despite the advanced feature extraction capabilities of deep learning architectures.

Notably, in Figure 15D,E, where the CNN-LSTM and CNN-LSTM-Attention models are compared, the introduction of the attention mechanism clearly improved recognition rates, pushing HB and OT accuracy to as high as 97%, and notably reducing confusion between RB and SM. This improvement suggests that attention helps the model better capture temporal dependencies and subtle differences in the vibration patterns, thus enhancing discrimination between similar shot types.

### 4.4. Smartphone Applications

We developed a user-friendly smartphone application that enables users to visualize the real-time recognition results captured by the sensor, providing an intuitive and enhanced viewing experience. As shown in Figure 17, this interface provides a real-time assessment of the RB’s goal-scoring status, as well as the acceleration and angular velocity values of the three axes at the moment of the goal. This visualization function allows users to monitor and analyze their shooting results in real time and provides professional insights.

## 5. Discussion

### 5.1. Application Scenarios

This study provides a scenario for exploring the application of artificial intelligence in sports. This research system has broad potential in future fields, such as smart stadiums, sports data analysis, and physical education. In smart stadiums, it can assist in the realization of unmanned scoring systems to automatically calculate goal scores. In the field of sports data analysis, it can automatically collect basketball game data, monitor athletes’ shooting movements in real time, and help coaches and analysts provide more targeted training suggestions. In physical education, teachers can more intuitively demonstrate the correct shooting posture and technical points to help athletes improve their skills. In short, the system promotes accurate and personalized services. Through these applications, not only are the efficiency and accuracy of sports competitions improved, but new tools are also provided for physical education and athlete training.

### 5.2. Limitations of This System

Our experimental results indicate that there is still room for improvement in the recognition accuracy of shooting outcomes. In future research, we plan to expand the sample size and enhance model performance by collecting data from a broader range of users and more varied scenarios. This is essential because real-world basketball environments and user groups are not limited to indoor courts or college students, as in our current study. Therefore, future studies should explore more complex and dynamic environments—such as outdoor courts, professional arenas, and amateur playgrounds—and include participants of different age groups, skill levels, and backgrounds to improve the system’s generalizability and adaptability.

Moreover, the accurate recognition of shooting outcomes is crucial not only for technical analysis but also for enhancing the user’s training experience and feedback quality. As the system is deployed in real-world environments, it is important to address potential issues such as long-term noise and sensor drift caused by positional shifts of the IMU sensors. In future work, we will incorporate a drift-aware learning mechanism, enabling the system to maintain robustness and accuracy under varying conditions of equipment placement and environmental change. Additionally, there is significant potential to improve the hardware platform. While our current design relies on IMU sensors, integrating complementary sensor modalities—such as pressure sensors, acoustic sensors, or vision systems (e.g., cameras)—at the basket and backboard could enrich the motion feature set and offer a more comprehensive analysis of shooting mechanics and outcomes. Nonetheless, we fully agree that battery life is a critical factor for broader deployment. In future iterations, we plan to explore higher-capacity batteries, intermittent computing techniques, and energy harvesting options to further extend system runtime while maintaining lightweight and wearable form factors. These enhancements will lay the groundwork for a more powerful and intelligent goal state recognition system.

Through the above improvements, we aim to develop a system that not only achieves higher recognition accuracy but also offers strong adaptability, scalability, and user-centered performance in diverse real-world training and competition settings.

## 6. Conclusions

This study provides a scenario for exploring the application of artificial intelligence in sports. This research system has broad potential in future fields, such as smart stadiums, sports data analysis, and physical education. In smart stadiums, it can assist in the realization of unmanned scoring systems to automatically calculate goal scores. In the field of sports data analysis, it can automatically collect basketball game data, monitor athletes’ shooting movements in real time, and help coaches and analysts provide more targeted training suggestions. In physical education, teachers can more intuitively demonstrate the correct shooting posture and technical points to help athletes improve their skills. In short, the system promotes accurate and personalized services.

Moreover, the method adopted in our study—integrating edge computing with deep learning models such as CNN, LSTM, and CNN-LSTM-Attention—offers several advantages. It enables real-time processing with low latency, ensures data privacy by minimizing cloud dependency, and enhances recognition accuracy even in complex movement scenarios. The modular structure of our system allows flexible deployment across different sports environments, while its low-power hardware ensures long-term operation. Through these applications and technical strengths, this system has the potential to significantly improve the efficiency and accuracy of sports competitions in the future, while also providing innovative tools for physical education and athlete training.

## Figures and Tables

**Figure 1 sensors-25-03709-f001:**
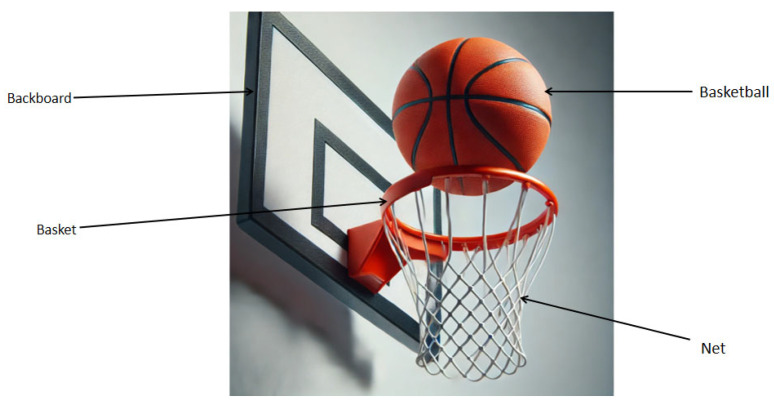
Schematic diagram of basketball components.

**Figure 2 sensors-25-03709-f002:**
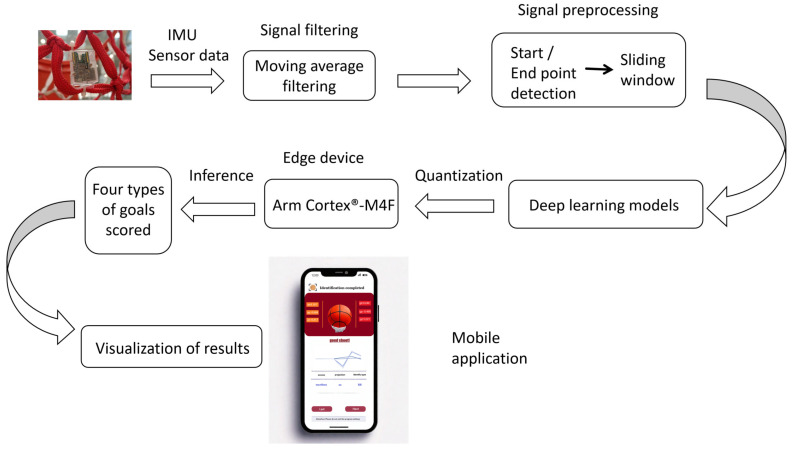
Overall system architecture.

**Figure 3 sensors-25-03709-f003:**
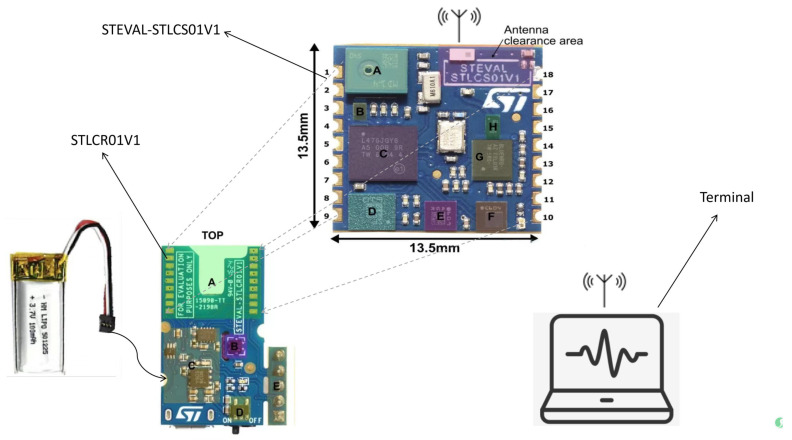
Schematic diagram of module connection. NOTE: “A: MP34DT04, MEMS digital microphones. B: LD39115J18R, 1.8V Low Quiescent Current Low Noise LDO. C: STM32L476JGY6, ARM Cortex-M4 32-bit microcontroller. D: LSM6DSM, Low-power 3-axis accelerometer and 3-axis gyroscope. E: LSM303AGR, Ultra-low power 3-axis accelerometer and 3-axis magnetometer. F: LPS22HB, MEMS pressure sensor: absolute digital output barometer with 260-1260 hPa. G: BlueNRG-MS, Bluetooth low-energy network processor. H: BALF-NRG-01D3, Balen integrated filter.”

**Figure 4 sensors-25-03709-f004:**
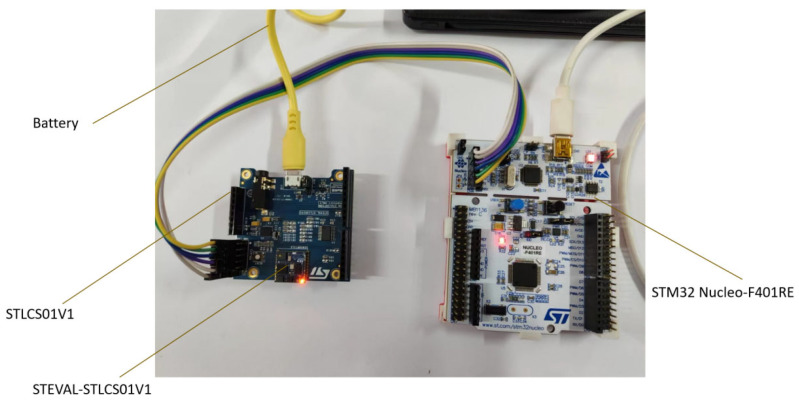
Development setup of Sensor Tile kit.

**Figure 5 sensors-25-03709-f005:**
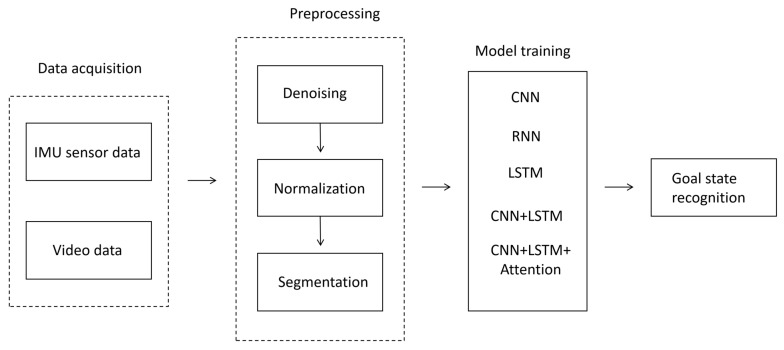
Experimental process.

**Figure 6 sensors-25-03709-f006:**
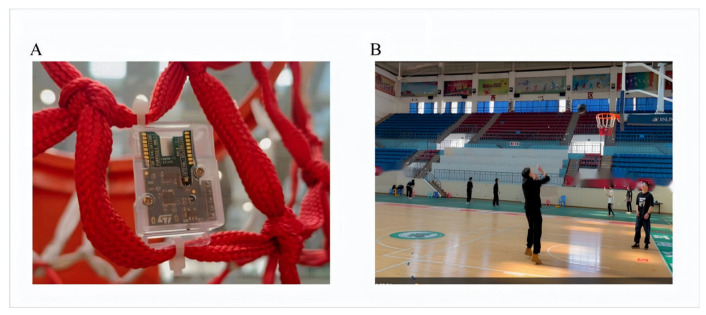
Experimental setup and data collection environment: (**A**) equipment installation and (**B**) data collection scenario.

**Figure 7 sensors-25-03709-f007:**
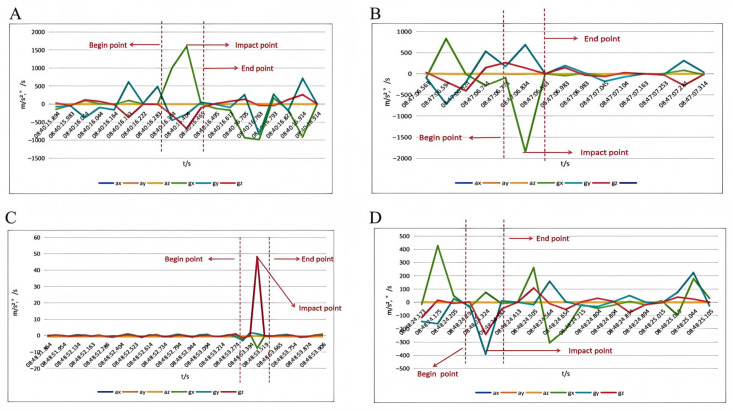
Temporal fluctuation patterns of different shot outcomes: (**A**) fluctuation in the rebounding sample; (**B**) fluctuation in the hollow sample; (**C**) no-goal shot fluctuations; and (**D**) variability in other goal states.

**Figure 8 sensors-25-03709-f008:**
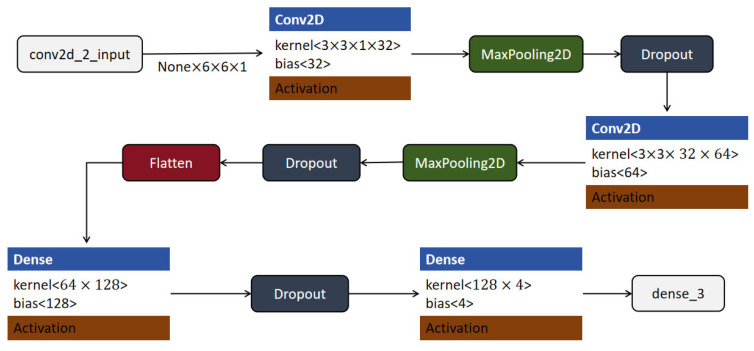
CNN model structure.

**Figure 9 sensors-25-03709-f009:**
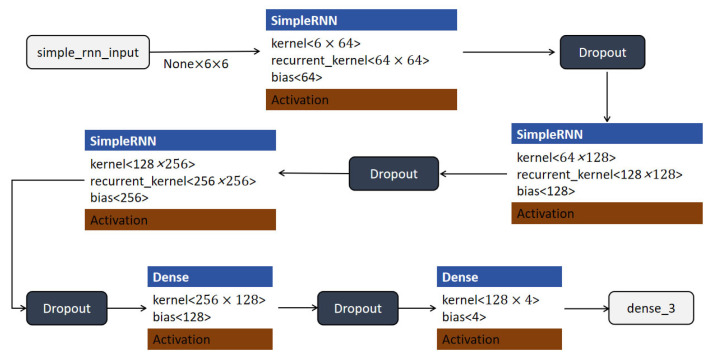
RNN model structure.

**Figure 10 sensors-25-03709-f010:**
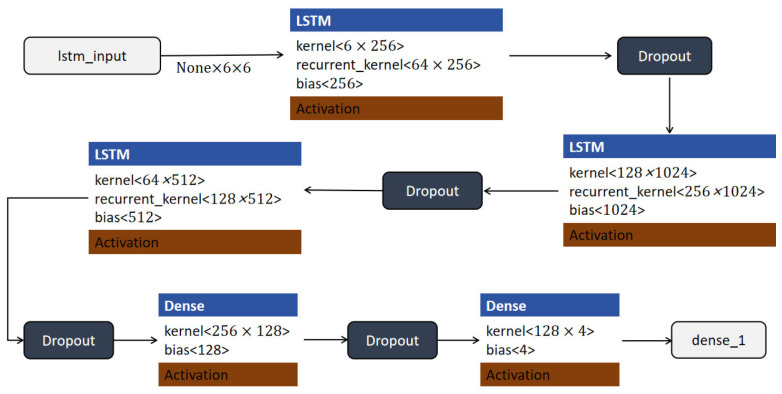
LSTM model structure.

**Figure 11 sensors-25-03709-f011:**
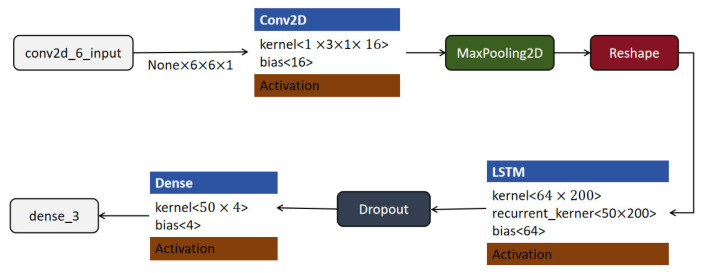
CNN-LSTM model structure.

**Figure 12 sensors-25-03709-f012:**
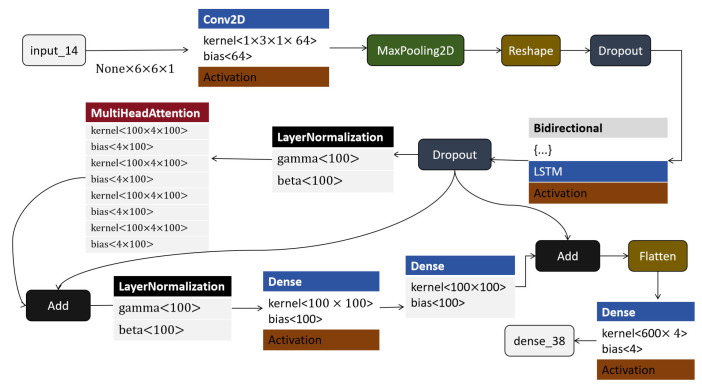
CNN-LSTM-Attention model structure.

**Figure 13 sensors-25-03709-f013:**
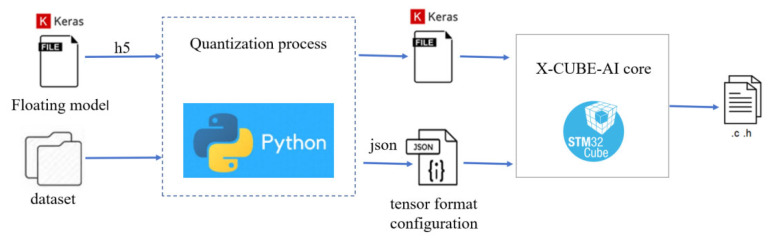
Deep learning model quantization and porting.

**Figure 14 sensors-25-03709-f014:**
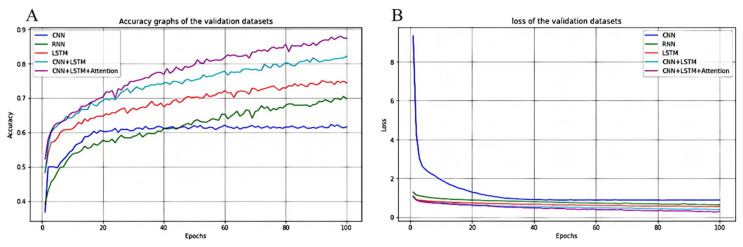
The performance of different models on the validation set during training: (**A**) verification dataset accuracy and (**B**) verification dataset loss value.

**Figure 15 sensors-25-03709-f015:**
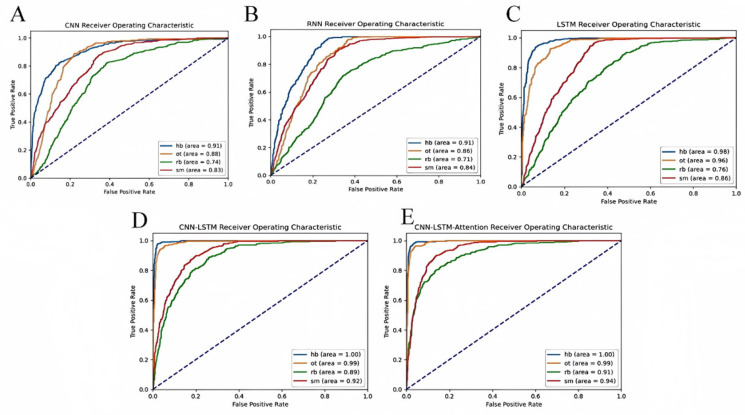
AUC-ROC curve comparison of five deep learning models: (**A**) AUC-ROC curve of CNN; (**B**) AUC-ROC curve of RNN; (**C**) AUC-ROC curve of LSTM; (**D**) AUC-ROC curve of CNN-LSTM; and (**E**) AUC-ROC curve of CNN-LSTM-Attention.

**Figure 16 sensors-25-03709-f016:**
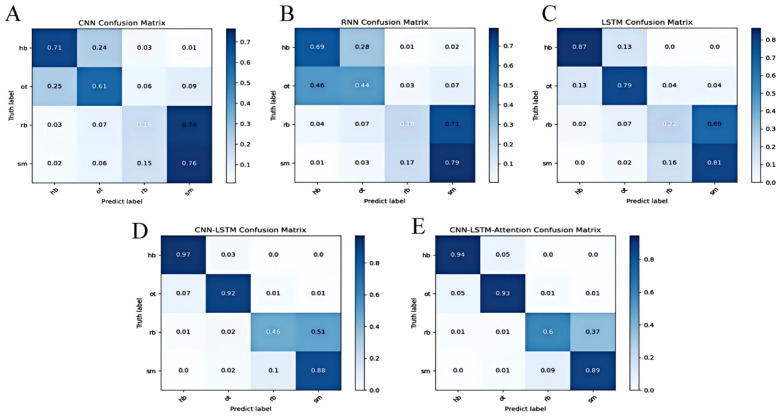
Confusion matrix comparison of five deep learning models: (**A**) CNN confusion matrix; (**B**) RNN confusion matrix; (**C**) LSTM confusion matrix; (**D**) CNN-LSTM confusion matrix; and (**E**) CNN-LSTM-Attention confusion matrix.

**Figure 17 sensors-25-03709-f017:**
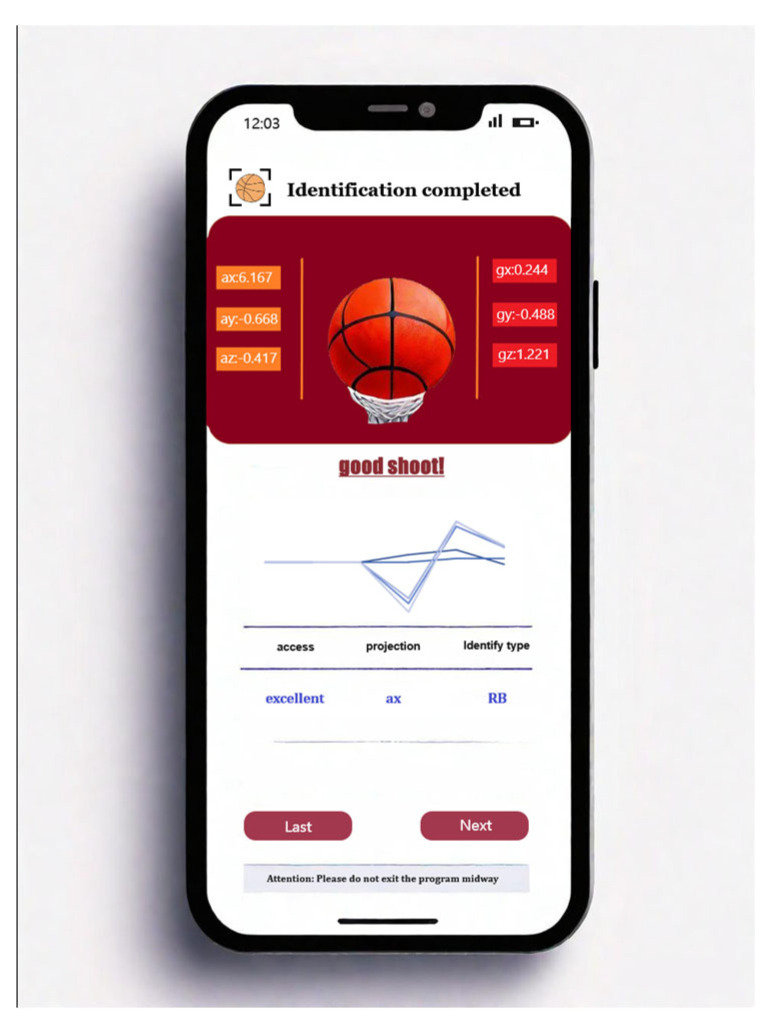
Screenshot of smartphone application.

**Table 1 sensors-25-03709-t001:** STEVAL-STLCS01V1 experimental settings.

Refer	Sensor Code	Usage
D	LSM6DSM	Low-power 3-axis accelerometer and 3-axis gyroscope
E	LSM303AGR	Ultra-low power 3-axis accelerometer and 3-axis magnetometer
G	Blue NRG-MS	Bluetooth low-energy network processor

**Table 2 sensors-25-03709-t002:** STLCR01V1 experimental settings.

Refer	Sensor Code	Usage
A	Sensor Tile footprint	Used to solder the Sensor Tile core system board
D	Power on/off switch	Power switch
E	SWD connector	5PIN SWD interface for programming and debugging

**Table 3 sensors-25-03709-t003:** Key specifications of the LSM6DSM inertial sensor used in the experiment. Note: The unit “√Hz” (square root Hertz) refers to spectral density. For example, “90 μg/√Hz” represents the acceleration noise density, which describes the noise amplitude per square root of bandwidth in hertz.

Sensor Component	Parameter	Specification	Configuration Used in Experiment
Accelerometer	Full-scale range	±2 g, ±4 g, ±8 g, ±16 g	±16 g
	Resolution	16-bit output data	16-bit
	Output data rate	Up to 6.6 kHz	416 Hz
	Sensitivity	~0.061 mg/LSB at ±2 g ~90 μg/√Hz	Corresponding sensitivity for ±16 g
	Noise density	~90 μg/√Hz	—
Gyroscope	Full-scale range	±125, ±245, ±500, ±1000, ±2000 dps	±2000 dps
	Resolution	16-bit output data	16-bit
	Output data rate	Up to 6.6 kHz	416 Hz
	Noise density	~4 mdps/√Hz	—

**Table 4 sensors-25-03709-t004:** Goal status labeling and classification based on vibration and video analysis.

Shot Status	Description	Vibration Pattern	Video Verification
Rebound (RB)	Ball hits the rebound and scores	Significant fluctuation	Yes
Hollow ball (HB)	Ball directly enters the basket	Fast and smooth fluctuation	Yes
Other (OT)	Complex cases, multiple rim touches	Irregular fluctuation	Yes
Shooting miss (SM)	Does not enter the hoop	Slight fluctuation	Yes

**Table 5 sensors-25-03709-t005:** Comparison of IMU-based activity recognition studies using machine learning and deep learning models.

Study	Sensor Type	Model Type	Task	Accuracy	Remarks
G Waltner et al., 2014 [50]	IMU	K-NN	Volleyball	77.56%	Dataset includes data from different participants, using simple machine learning
W Gomaa et al., 2017 [51]	IMU	RF	Human activities	80%	Dataset includes 14 activities
Hoelzemann et al., 2023 [52]	IMU	CNN, LSTM	Basketball activity recognition	85%	Dataset includes diverse players; baseline models evaluated
Xiaoyu Guo et al., 2023 [53]	IMU	CNN	Basketball activity recognition	82%	The datasets come from participants of different levels
Our study	IMU	CNN, RNN, LSTM	Basketball state recognition	87.79%	Datasets from different participants; deep learning models used

**Table 6 sensors-25-03709-t006:** Comparison of different algorithm models.

Model	Average Accuracy	Number of Parameters	Recall	Precision	F1 Score	Maximum Latency
CNN	64.96%	93,184	61.03%	55.33%	54.63%	3 ms
RNN	56.83%	48,042	52.58%	51.99%	51.25%	2 ms
LSTM	68.85%	69,636	67.02%	66.72%	65.9%	3 ms
CNN-LSTM	81.14%	232,680	80.47%	82.22%	80.49%	3 ms
CNN-LSTM-Attention	87.79%	307,360	87.35%	88.69%	87.65%	6 ms

## Data Availability

Data is contained within the article.

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
