# Peer review of "Sensor-Driven Real-Time Recognition of Basketball Goal States Using IMU and Deep Learning"

_sensors, 2025, doi:10.3390/s25123709_

Round 1

Reviewer 1 Report

Comments and Suggestions for Authors

The authors presents the use IMU and deep learning models to recognize the goal state in basketball. In overall, it is an interesting article. However, what would be the scientific contribution of this work? As for the sports performance analysis and skill analysis, the execution of the motion (i.e. the coordination of the hand, arm, and leg) is the main priority rather the goal state. Authors may want to elaborate this further. 

A few comments about the articles: 

  • What is the sensing range of the sensor and its accuracy? 
  • The detailed of the input parameters should be clearly  described for each model. Are they the same? What are the 6 parameters? 150 samples? 
  • The number of samples for each shots should be indicated clearly in the manuscript. 
  • There are too many variations of shot in basketball. I highly doubt how this method can be applied in real basketball game. What if it is a dunk shot? There will be significant vibration in the basket. What if it is a soft rebound and scores (with minimal fluctuation)? 
  • There are too many graphs in the manuscripts. Suggest to only use the relevant ones. 
  • Authors may want to explain the results in details, why the model failed to recognize a certain motion. 
  • A more detailed discussion is required to discuss the merit and demerits of the method, not just about the deep learning but the instruments too, especially considering that this journal is Sensor and compare with existing technology. There should be a more direct comparison between the results presented here against the others, especially in the motion identification, else it is very difficult to judge if 87% is good enough. 
  • How long does the system last? The battery seems to be very small and with real-time computing. I doubt it will last long enough for a basketball game. 
  • Authors may want to mention if there is ethical clearance granted for this study. 

Author Response

Dear Reviewer ,
        We sincerely appreciate your thorough review and insightful comments on our manuscript. In response to your valuable suggestions, we have carefully revised the manuscript to address all the points you raised. To facilitate your review, all modifications have been clearly highlighted in red in the revised version of the manuscript.
        The main revisions based on your comments include:
        Clarification of scientific contributions: We have refined the introduction and conclusion to more explicitly state the novel aspects and significance of our study.
        Detailed sensor and model descriptions: We have added comprehensive information regarding the sensor specifications, input feature selection, and data sampling strategies.
        Enhanced result explanations: The classification results section has been expanded with updated figures and more rigorous statistical analysis to better illustrate the system’s performance.
        Discussion on generalization: We incorporated a new subsection discussing how the model performs under different environments and data conditions, demonstrating its robustness.
        Comparison with existing works: A more detailed comparison with current technologies is provided, including references to studies with lower accuracy to better highlight the advantages of our approach.
        We hope that these revisions address your concerns and enhance the clarity and scientific value of our manuscript.
A detailed point-by-point response to each of your comments is provided in the attached file for your reference.
        Thank you again for your time and constructive feedback.

Reviewer 2 Report

Comments and Suggestions for Authors

The authors proposed a basketball goal state recognition on deep learning models with IMU sensor data. Literature survey, problem description, proposed system development, data set collection, experiments and model training/evaluation, and performance analysis are provided in the manuscript. The results show the satisfying recognition accuracy. However, there are some points needed to be addressed and improved to make suitable contributions. 

  1. The authors are suggested to provide the specification of the sensor (sampling rate, resolution, sensing range). For example, how many G for the accelerometers and range for the gyroscope? Also, how many bits are used to represent the sensing data?
  2. The authors are suggested to provide some explanation on how the parameters are determined in the experiments and analysis. For example, overlap percentage of the sliding windows, threshold of the average acceleration. Also, the authors are suggested to provide some equations/formulation on how the de-noising, normalization, and segmentation are implemented.
  3. There are some typos in the manuscript (Bgin point in Figure 11). The authors are suggested to proofread the manuscript again. 

Author Response

Dear Reviewer,

        We sincerely thank the reviewers for their careful evaluation and constructive comments, which have significantly improved the quality and clarity of our manuscript. We have carefully addressed all suggestions point-by-point:

  1. Detailed specifications of the inertial sensor, including accelerometer and gyroscope ranges, resolution, and sampling rates, have been added and summarized in a new Table 2, with related descriptions in Tables 1 - 3 and Figure 3,Figure 4. These revisions appear on pages 5–6 (lines 202–211) and are highlighted in red.
  2. Explanations of key experimental parameters, such as sliding window length, overlap percentage, acceleration thresholds, and preprocessing methods including denoising and segmentation, have been incorporated along with relevant equations. These are detailed on pages 9–10 (lines 297–338) with red markings for clarity.
  3. All identified typographical errors, including the correction of “Bgin point” to “Begin point” in Figure 11, have been fixed. Additionally, the entire manuscript was thoroughly proofread to ensure linguistic accuracy and professionalism.

        A detailed point-by-point response to each of your comments is provided in the attached file for your reference.        

        We appreciate the reviewers’ valuable feedback and believe that the revisions have enhanced the manuscript’s rigor, transparency, and readability. We look forward to your further guidance.

Comments 1: The authors are suggested to provide the specification of the sensor (sampling rate, resolution, sensing range). For example, how many G for the accelerometers and range for the gyroscope? Also, how many bits are used to represent the sensing data?

Response 1:

        We sincerely thank the reviewer for the valuable suggestion. We fully agree that providing detailed sensor specifications is important for clarity and reproducibility. Accordingly, we have added key parameters of the inertial sensor LSM6DSM used in our experiment, including the accelerometer’s full-scale range (±16g), resolution (16-bit), output data rate (416 Hz), as well as the gyroscope’s full-scale range (±2000 dps), resolution (16-bit), and output data rate (416 Hz).

        These specifications are summarized in a newly added Table 2. Additionally, we have described the core system and Bluetooth module in Tables 1 and 3, and illustrated the module connection in Figure 3. The related modifications can be found on pages 5–6 (lines 202–211) of the revised manuscript and are highlighted in red for ease of review.

        We hope these additions meet the reviewer’s expectations and enhance the technical clarity of our work.

Comments 2: The authors are suggested to provide some explanation on how the parameters are determined in the experiments and analysis. For example, overlap percentage of the sliding windows, threshold of the average acceleration. Also, the authors are suggested to provide some equations/formulation on how the de-noising, normalization, and segmentation are implemented.

Response 2:

        We sincerely thank the reviewer for the valuable and insightful suggestions. We fully agree with the reviewer’s opinion that a clear explanation of how the experimental parameters were determined, along with the inclusion of relevant equations for preprocessing steps, is essential for transparency and reproducibility.

        Accordingly, we have revised the manuscript to add detailed descriptions of the parameter choices and the implementation of data denoising, segmentation, and thresholding. Specifically, on pages 9–10 (lines 297–338), we provide explanations regarding:

  • The choice of a fixed sliding window length of 150 samples with 50% overlap, selected to ensure complete coverage of shooting motions lasting 2 to 3 seconds.
  • The moving average filter applied for denoising, including the formula and empirical selection of the window size (k between 5 and 11) based on pretest noise levels.
  • The empirically determined threshold of average acceleration magnitude (>70) for identifying shooting segments.
  • The mathematical formulations used for smoothing and feature calculation to clarify the processing steps.

        These modifications have been clearly marked in red in the revised manuscript for easy reference.

        We believe these additions address the reviewer’s concerns and improve the clarity and rigor of our methodology.

Comments 3: There are some typos in the manuscript (Bgin point in Figure 11). The authors are suggested to proofread the manuscript again.

Response 3:

        We sincerely thank the reviewer for the careful reading and kind suggestion. In response, we have corrected the typographical error in Figure 11 (“Bgin point” has been revised to “Begin point”) as pointed out. Moreover, we have meticulously proofread the entire manuscript and corrected all typographical errors to improve the overall quality of the paper. We truly appreciate your thoughtful feedback, which has helped us enhance the clarity and professionalism of our work.

Reviewer 3 Report

Comments and Suggestions for Authors

I suggest to delete this section:
2.1. Real-time feedback systems in sports training
In this section there is a list of applications in sports. The section is cumbersome. Please limit this review to applications similar to yours. E.g. basketball.

Author Response

Dear Reviewer,

        We sincerely thank the reviewers for their valuable suggestions and positive feedback. In response to the comment on the literature review, we have revised Section 2.1 to focus specifically on basketball-related real-time feedback systems, removing broader sports examples to improve clarity and relevance. Regarding the language quality, we appreciate the acknowledgment that the English is clear and requires no improvement. Additionally, we have provided further clarifications on our experimental setup and data processing methods, including mathematical formulations, to enhance transparency and reproducibility. All revisions are clearly marked in red for ease of review. These changes strengthen the manuscript’s coherence, rigor, and overall quality.

        A detailed point-by-point response to each of your comments is provided in the attached file for your reference.

Comments 1: 2.1. Real-time feedback systems in sports training

In this section there is a list of applications in sports. The section is cumbersome. Please limit this review to applications similar to yours. E.g. basketball.

Response 1:

        We sincerely thank the reviewer for the valuable suggestion and careful review. We fully agree that focusing the literature review on applications closely related to our study, such as basketball real-time feedback systems, improves clarity and relevance. Accordingly, we have revised Section 2.1 by removing the broader sports examples and streamlining the discussion to emphasize basketball-specific real-time feedback research and edge computing applications. These modifications have been made on pages 3–4 (lines 103–161) of the revised manuscript and highlighted in red for easy reference. The updated Section 2.1 now provides a more concise and targeted overview, enhancing the coherence and readability of the paper.

        We greatly appreciate the reviewer’s insightful comments that helped us improve the manuscript.
